# The low-density lipoprotein receptor promotes infection of multiple encephalitic alphaviruses

Hongming Ma [1], Lucas J. Adams [2], Saravanan Raju[1,2], Alan Sariol[1], Natasha M. Kafai[1], Hana Janova[1,2], William B. Klimstra[3], Daved H. Fremont [2,4,5] & Michael S. Diamond [1,2,5,6,7] ✉

Members of the low-density lipoprotein receptor (LDLR) family, including LDLRAD3, VLDLR, and ApoER2, were recently described as entry factors for different alphaviruses. However, based on studies with gene edited cells and knockout mice, blockade or abrogation of these receptors does not fully inhibit alphavirus infection, indicating the existence of additional uncharacterized entry factors. Here, we perform a CRISPR-Cas9 genome-wide loss-of-function screen in mouse neuronal cells with a chimeric alphavirus expressing the Eastern equine encephalitis virus (EEEV) structural proteins and identify LDLR as a candidate receptor. Expression of LDLR on the surface of neuronal or non-neuronal cells facilitates binding and infection of EEEV, Western equine encephalitis virus, and Semliki Forest virus. Domain mapping and binding studies reveal a low-affinity interaction with LA domain 3 (LA3) that can be enhanced by concatenation of LA3 repeats. Soluble decoy proteins with multiple LA3 repeats inhibit EEEV infection in cell culture and in mice. Our results establish LDLR as a low-affinity receptor for multiple alphaviruses and highlight a possible path for developing inhibitors that could mitigate infection and disease.

Alphaviruses are mosquito-transmitted RNA viruses that can infect a range of vertebrate hosts, including humans, non-human primates, horses, rodents, and birds. Alphaviruses historically have been categorized by geographical location into "Old World" or "New World" viruses, although their global spread over the past several decades has blurred such definitions[1,2]. Old World alphaviruses include chikungunya (CHIKV), O'nyong-nyong (ONNV), Ross River (RRV), Mayaro (MAYV), Semliki Forest (SFV), and Sindbis (SINV) viruses, and generally cause acute debilitating arthritis that can lead to chronic musculoskeletal disease in some patients[3]. New World alphaviruses include Eastern (EEEV), Venezuelan (VEEV), and Western (WEEV) equine encephalitis viruses and cause a febrile syndrome that can rapidly progress to invasion of the brain and spinal cord, resulting in neurological disease and mortality[4–6]. Though naturally transmitted by mosquitoes, encephalitic alphaviruses have been weaponized in the past and can be spread through an aerosol route with a high degree of virulence[7,8]. Currently, there are no approved vaccines or therapies for any alphavirus infection.

The alphavirus RNA genome encodes four nonstructural proteins (nsP1, nsP2, nsP3, and nsP4), which mediate viral translation, replication,

[1]Department of Medicine, Washington University School of Medicine, St. Louis, MO 63110, USA. [2]Department of Pathology & Immunology, Washington University School of Medicine, St. Louis, MO 63110, USA. [3]The Center for Vaccine Research and Department of Immunology, The University of Pittsburgh, Pittsburgh, PA 15261, USA. [4]Department of Biochemistry & Molecular Biophysics, Washington University School of Medicine, St. Louis, MO 63110, USA. [5]Department of Molecular Microbiology, Washington University School of Medicine, St. Louis, MO 63110, USA. [6]Andrew M. and Jane M. Bursky Center for Human Immunology and Immunotherapy Programs, Washington University School of Medicine, St. Louis, MO 63110, USA. [7]Center for Vaccines and Immunity to Microbial Pathogens, Washington University School of Medicine, Saint Louis, MO 63110, USA. ✉e-mail: mdiamond@wustl.edu

and host immune evasion, and six structural proteins (capsid, E3, E2, 6K, transframe [TF], and E1). The E2 and E1 structural proteins form a heterodimer and assemble into 80 trimeric spikes on the alphavirus surface arranged with $T = 4$ icosahedral symmetry[9]. Within each E1-E2 heterodimer, the E2 glycoprotein is preferentially exposed and shields the majority of the E1 protein from solvent, including the conserved fusion loop required for penetration into the cytoplasm[10]. Alphaviruses are internalized into cells principally by clathrin-mediated endocytosis, and membrane fusion occurs in endosomes[11,12]. Upon fusion, alphavirus particles disassemble, releasing genomic RNA into the cytoplasm of infected cells. The viral genome is translated, which facilitates negative- and positive-strand RNA synthesis and viral replication. Nascent virions are formed by budding from the host cell plasma membrane[13].

Engagement with host receptors enables alphavirus entry into cells[14,15]. Over the last few years, genetic screens have identified several alphavirus receptors, and many of these interactions have been corroborated by biochemical, biophysical, and high-resolution cryo-electron microscopy studies. For example, an RNA interference (RNAi) screen identified NRAMP/NRAMP2 as a receptor for SINV in *Drosophila* and mammalian cells[16]. CRISPR-Cas9 screens identified the immunoglobulin (Ig)-like domain containing molecule MXRA8 as an entry receptor for several members of the Semliki Forest (SF) complex including CHIKV, RRV, MAYV, ONNV, and Getah-like viruses[17–22]. LDLRAD3, a member of the low-density lipoprotein receptor (LDLR) family, was defined as a key entry receptor for VEEV that mediates pathogenesis in mice[23–25]. VLDLR and ApoER2, two related LDLR family proteins, were identified as receptors for EEEV, SFV, and to a lesser extent SINV[26,27].

As NRAMP2, MXRA8, LDLRAD3, VLDLR, and ApoER2 are not used by all alphaviruses, and blockade or loss of expression of some of these proteins does not abrogate virus infection, we postulated the existence of additional entry factors that could mediate infection, possibly in a cell-type specific manner[15]. Here, we performed a CRISPR-Cas9 genome-wide screen to identify additional receptors for encephalitic alphaviruses in murine neuronal cells. We show that LDLR can act as a low-affinity entry factor for EEEV, WEEV, and to a lesser extent SFV. Mapping studies identified a dominant mode of EEEV binding with the LA3 domain of LDLR. Using this information, we designed soluble LA3 domain-based LDLR receptor decoys that inhibit alphavirus infections in cell culture and in mice.

## Results

### A CRISPR/Cas9 genome-wide screen identifies LDLR as a host factor that promotes EEEV infection

To identify entry factors for EEEV, we used a CRISPR/Cas9 loss-of-function genome-wide screening approach. The GeCKO v2 library[28] was constructed in Neuro2a (N2a) murine neuronal cells; this cell line was used because neurons are targets of EEEV infection in vivo[29]. To reduce background associated with attachment promoted by heparan sulfate molecules[30,31], we used a N2a cell clone with a gene deficiency of *B4galt7*[23], which encodes a key enzyme in the glycosaminoglycan synthesis pathway. We performed the screen by inoculating the library-containing N2a *B4galt7* cells with SINV-EEEV, a chimeric virus that encodes the nonstructural genes (nsP1, nsP2, nsP3 and nsP4) of SINV and the structural genes (C, E3, E2, 6K and E1) of EEEV strain FL93-939 (a select agent strain), along with a green fluorescent reporter gene (GFP) (Supplementary Fig 1a), which enabled experiments to be performed at BSL2 and under non-select agent security conditions since chimeric alphaviruses are attenuated in immunocompetent animals[32,33]. Under high multiplicity-of-infection conditions (MOI of 5), virtually all cells die within 4 days. The few library-containing cells that survived the infection were collected, propagated, and then re-inoculated with SINV-EEEV. After three rounds of infection and propagation, genomic DNA from the surviving cells was collected, and single-guide (sg)RNAs were sequenced and analyzed (Fig. 1a **and** Supplementary Data 1).

To focus on potential entry factors for EEEV, we picked the top five candidates (*e.g.*, *Sema5a*, *Olfr970*, *Ldlr*, *Ccp110*, and *Svs2*) that are known or predicted to be expressed on the plasma membrane for validation (Supplementary Fig 1b). N2a *B4galt7* cells in bulk were subjected to gene editing with 2 to 3 independent sgRNAs. Of the five candidate genes tested, a loss-of-infection phenotype was confirmed only with sgRNA targeting *Ldlr*. To corroborate these results, we generated *ΔLdlr* single-cell clones in N2a *B4galt7* cells and complemented them with full-length *Ldlr* cDNA encoding mutations in the sgRNA seed-sequence (Supplementary Fig 1c, d). A deficiency or complementation of LDLR did not affect cell viability (Supplementary Fig 1e). However, we observed marked reductions of SINV-EEEV infection in N2a *ΔB4galt7 ΔLdlr* cells compared to N2a *ΔB4galt7* control sgRNA cells under single- and multi-step growth conditions (Fig. 1b, c). Complementation of N2a *ΔB4galt7 ΔLdlr* cells with *Ldlr* restored LDLR expression and SINV-EEEV infectivity (Fig. 1c and Supplementary Fig 1d). We confirmed the relationship between LDLR expression and SINV-EEEV infectivity using a human THP-1 cell line that has very low surface expression of LDLR and was relatively resistant to infection with SINV-EEEV at baseline (Fig. 1d and Supplementary Fig 1f). Ectopic expression of LDLR in THP-1 cells resulted in enhanced SINV-EEEV infection (Fig. 1d and Supplementary Fig 1f).

Because SINV-EEEV is a chimeric virus, we validated our results with multiple strains of Madariaga virus (MADV), which comprise a non-select agent lineage of EEEV that circulates in South America and principally infects animal reservoirs but also can cause infection and disease in humans[34,35]. Five MADV strains (Argentina 1936, Peru 1970, Brazil 1975, Brazil 1985, and Colombia 1992; Supplementary Table 1) were selected from the three existing lineages. As seen with SINV-EEEV, all five MADV strains showed reduced infection in N2a *ΔB4galt7 ΔLdlr* cells under single-step growth conditions (Supplementary Fig 1g), and this result was confirmed with one strain (Argentina 1936) under multi-step growth conditions (Fig. 1e). Complementation of N2a *ΔB4galt7 ΔLdlr* cells with LDLR restored infectivity of all MADV strains (Fig. 1e and Supplementary Fig 1g).

Previous studies have shown that alphaviruses within and across family complexes can use overlapping sets of alphavirus receptors (*e.g.*, MXRA8 and VLDLR)[18,26] for entry and infection. Accordingly, we tested whether LDLR expression in N2a cells affected infectivity of other alphaviruses, including chimeric or authentic viruses in the VEE (SINV-VEEV), WEE (SINV and SINV-WEEV), and SF (SINV-CHIKV, SINV-SFV, MAYV, and Getah virus [GETV]) complexes. Whereas infection with SINV-WEEV, SINV-SFV, and GETV was decreased in N2a *ΔB4galt7 ΔLdlr* cells, SINV, SINV-CHIKV, MAYV, and SINV-VEEV did not show significantly reduced infection in N2a *ΔB4galt7 ΔLdlr* cells compared to N2a *ΔB4galt7* cells (Fig. 1b, c, Supplementary 1h), suggesting that for some alphaviruses, LDLR is not required for infectivity in these cells, possibly due to expression of alternate entry factors. For SINV-WEEV, complementation of N2a *ΔB4galt7 ΔLdlr* cells with *Ldlr* restored infection under multi-step growth conditions (Fig. 1c). Enhanced infectivity also was confirmed for SINV-WEEV and SINV-SFV in THP-1 cells ectopically expressing LDLR (Fig. 1f, g).

### LDLR modulates SINV-EEEV attachment and internalization

Although expression of LDLR facilitated SINV-EEEV infection, the loss-of-function and complementation experiments did not establish its requirement for entry. To address this question, we performed binding and internalization assays with LDLR-expressing THP-1 cells. SINV-EEEV showed increased binding at 4 °C after 90 min to THP-1 cells expressing LDLR compared to control cells that lacked expression (Fig. 2a). When virus internalization assays were performed at 37 °C for 1 h, greater amounts of SINV-EEEV, as judged by intracellular viral RNA levels, were present within LDLR-expressing cells than in control THP-1 cells (Fig. 2b). We similarly observed greater SINV-EEEV binding and

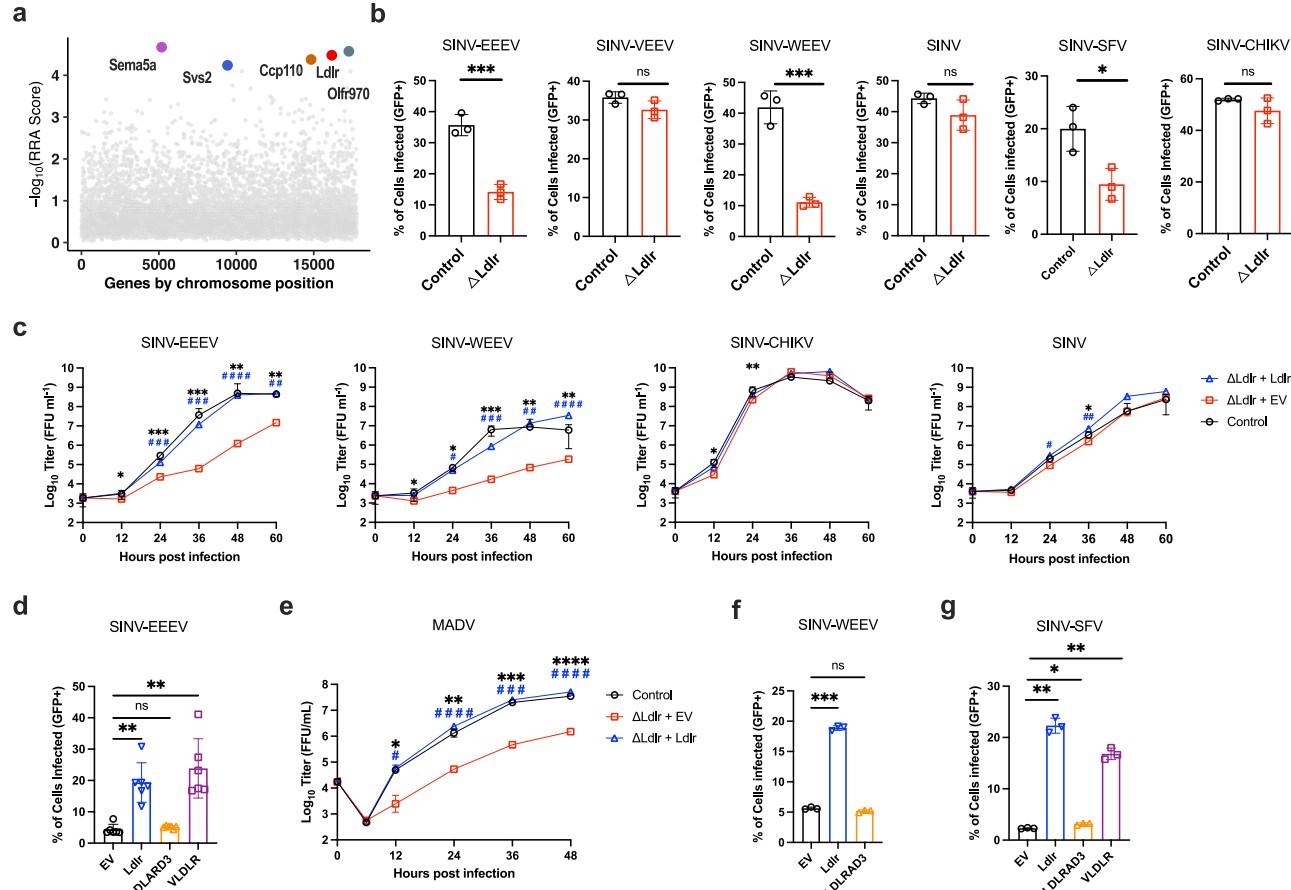

**Fig. 1 | LDLR promotes infection of EEEV and other alphaviruses in cells.**
**a** Results of CRISPR-Cas9 genome wide loss-of-function SINV-EEEV infection screen in *ΔB4galt7* N2a cells. Enriched genes based on robust rank aggregation (RRA) scores in the SINV-EEEV-selected population. **b** *ΔB4galt7* (Control) or *ΔB4galt7 ΔLdlr* (*ΔLdlr*) N2a cells were inoculated with SINV-EEEV-GFP (FL93-939), SINV-VEEV-GFP (TrD), SINV-WEEV-GFP (CBA87), SINV-GFP (TR339), SINV-SFV-GFP (SFV4), or SINV-CHIKV-GFP (LR2006) for 6.5 to 24 h (depending on the virus), and infection was assessed by flow cytometry for GFP expression (*n* = 3 experiments). **c** Multistep growth curves of SINV-EEEV, SINV-WEEV, SINV-CHIKV, and SINV in *ΔB4galt7* (Control), *ΔB4galt7 ΔLdlr* (*ΔLdlr* + EV), and LDLR-complemented *ΔB4galt7 ΔLdlr* (*ΔLdlr* + *Ldlr*) N2a cells (*n* = 3 experiments). **d** Human THP-1 cells ectopically expressing empty vector (EV), *Ldlr*, *LDLRAD3*, and *VLDLR* were inoculated with SINV-EEEV-GFP, and infection was assessed by flow cytometry (*n* = 6 experiments). **e** Multistep growth curves of MADV (Argentina 1936) in *ΔB4galt7* (Control), *ΔB4galt7 ΔLdlr* (*ΔLdlr* + EV) and complemented *ΔB4galt7 ΔLdlr* (*ΔLdlr* + *Ldlr*) N2a cells (*n* = 3 experiments). Black asterisks indicate comparisons between *ΔB4galt7* and *ΔB4galt7*

*ΔLdlr* N2a cells. Blue pound signs indicate comparisons between *ΔB4galt7 ΔLdlrad3* and LDLR-complemented *ΔB4galt7 ΔLdlrad3* N2a cells. **f, g** Human THP-1 cells ectopically expressing empty vector (EV), *Ldlr*, *LDLRAD3*, or *VLDLR* were inoculated with SINV-WEEV-GFP (**f**) or SINV-SFV-GFP (**g**), and infection was assessed by flow cytometry (*n* = 3 experiments). Means ± SD are shown. Statistical analysis (*P* values from left to right): (**b**) two-tailed unpaired t test: ****P* = 0.0009, ****P* = 0.0007, **P* = 0.0251; (**c, e**) two-way ANOVA with Dunnett's post-test: **P* = 0.0269, ****P* = 0.0002, ****P* = 0.0003, ***P* = 0.0055, ***P* = 0.0075, ###*P* = 0.0007, ###*P* = 0.0004, ####*P* < 0.0001, ##*P* = 0.0036 (**c**, SINV-EEEV); **P* = 0.0467, **P* = 0.0178, ****P* < 0.0001, ***P* = 0.0066, ***P* = 0.0017, #*P* = 0.0168, ###*P* = 0.0003, ##*P* = 0.0017, ####*P* < 0.0001 (**c**, SINV-WEEV); **P* = 0.0461, ***P* = 0.0053 (**c**, SINV-CHIKV); **P* = 0.0244, #*P* = 0.0186, ##*P* = 0.0024 (**c**, SINV); **P* = 0.0280, ***P* = 0.0038, ****P* = 0.0004, *****P* < 0.0001, ***P* = 0.0075, #*P* = 0.0170, ####*P* < 0.0001, ###*P* = 0.0005, ####*P* < 0.0001 (**e**); (**d, f, g**) one-way ANOVA with Dunnett's post-test: ***P* = 0.0092, ***P* = 0.0077 (**d**); ****P* = 0.0002 (**f**); ***P* = 0.0031, **P* = 0.0388, ***P* = 0.0034 (**g**); ns, not significant. Source data are provided as a Source Data file.

internalization in N2a neuronal cells expressing LDLR compared to LDLR-deficient cells (Supplementary Fig 2a, b).

## LDLR directly binds to EEEV
LDLR is a type I membrane protein with an ectodomain composed of seven N-terminal LDL-receptor class-A (LA) repeats, two EGF-like domains, a beta-propeller domain, another EGF-like domain, and a stretch of residues with multiple *O*-link glycosylation sites (Fig. 3a). As the LA domains mediate binding of endogenous low-density lipoprotein ligands[36], they are collectively termed the ligand binding domain (LBD). The LBD of LDLR also contains the binding site for VSV and human rhinovirus subtype B, and the LBD of VLDLR mediates binding to EEEV and SFV[26,27]. As such, we evaluated the direct binding activity of LDLR-Fc (full ectodomain fused to human IgG1 Fc) to EEEV virus-like particles (VLPs: C, E2, and E1)[37] using biolayer interferometry (BLI). EEEV VLPs in solution bound to captured LDLR-Fc but not to LDLRAD3-D1-Fc, which instead bound to VEEV (Fig. 3b).

Recent studies with VEEV and LDLRAD3[23–25] and SFV and VLDLR[27] suggested that individual LA domains can engage alphavirus structural proteins. To determine whether specific LA domains bind to EEEV, we engineered single LA repeat domains as fusion proteins with human IgG1 Fc (Supplementary Fig 3a). Binding experiments by BLI were performed with each single LA domain Fc fusion protein and EEEV VLPs. Notably, EEEV VLPs bound directly to LA3-Fc (Fig. 3c), but not to LA1-Fc, LA-2-Fc, LA4-Fc, LA5-Fc, LA6-Fc, or LA7-Fc. This experiment was performed with LDLR LA domains immobilized on the BLI biosensor such that an individual VLP could interact with multiple adjacent proteins to enhance avidity, akin to how LDLR clusters on the cell surface (Fig. 3e and Supplementary Fig 3b). However, when the assay design was reversed, with the EEEV VLP captured by monoclonal antibodies (mAbs), binding with LDLR domain LA3, LDLR-Fc, or LBD-Fc in solution was negligible suggesting a low-affinity interaction (Figs. 3d and 4c).

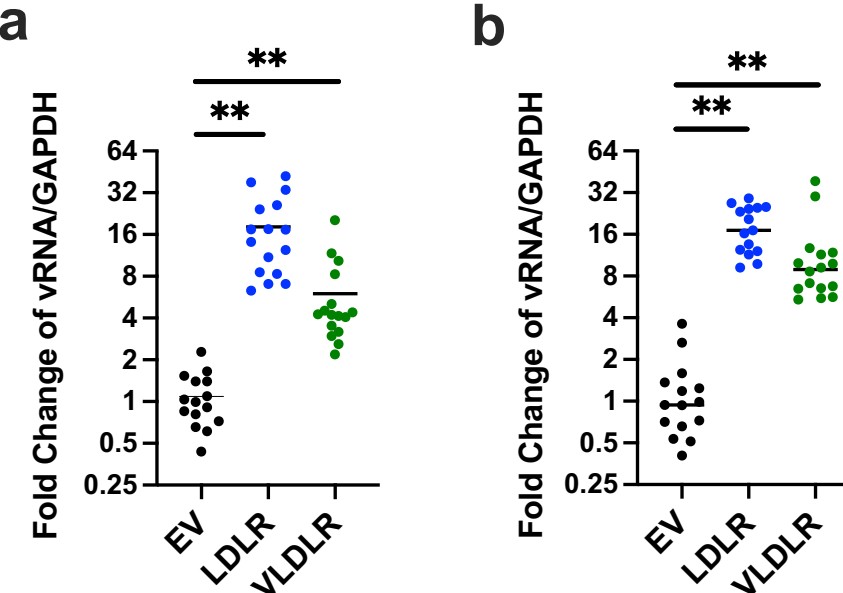

**Fig. 2 | LDLR expression promotes SINV-EEEV attachment and internalization.** **a** SINV-EEEV (FL93-939) was incubated with THP-1 cells expressing empty vector (EV), LDLR, or VLDLR at 4 °C for 90 min. After extensive washing, bound virions were quantified as the ratio of viral RNA (vRNA) to *GAPDH* gene mRNA level via RT-qPCR and then normalized to the control EV cells. **b** After removal of unbound virus, the temperature was increased to 37 °C for 1 h to allow internalization. After proteinase K digestion to remove surface-associated virus, intracellular viral RNA was measured and presented as in (**a**). Mean ± SD (*n* = 4 experiments in quadruplicate, all data points shown). Statistical analysis (*P* values from left to right) on mean values of four experiments: one-way ANOVA with Dunnett's post-test: (**a**) **P = 0.0024, **P = 0.0039; (**b**) **P = 0.0013, **P = 0.0043. Source data are provided as a Source Data file.

## Mapping of the LDLR LA3 domain binding site for EEEV

To gain insight as to the LA domain binding preference, we mapped interaction residues in the LA3 domain using structure-guided mutagenesis and infection assays. We chose the LA3 domain because it was the only one that showed avid binding to EEEV VLPs (Fig. 3c). The LA domain is a conserved module of approximately 40 amino acids with six cysteines forming three disulfide bonds and a calcium ion coordinated by acidic residues near the C-terminus (Supplementary Fig 3c). We first engineered a "mini-receptor" by appending the LA3 domain to the N-terminus of the membrane proximal *O*-link glycosylation site region (LA3-OL; Fig. 3a). Expression of LA3-OL with an N-terminal FLAG tag (Supplementary Fig 3d) enhanced EEEV infection in THP-1 cells, although to a lesser degree than the full-length LDLR (Fig. 3f). Mutations were designed using a BLOSUM scoring matrix[38] and introduced into the FLAG-LA3-OL construct for transduction into THP-1 cells. Cells that efficiently expressed LA3 mutants on their cell surface (Supplementary Fig 3e) were inoculated with SINV-EEEV (Fig. 3g). We identified three amino acid groups (Asp112 and Glu113, Phe126 and Val127, and Asp136 to Ala141) that significantly impacted the ability of the LA3 domain to support SINV-EEEV infection, with the Asp136 to Ala141 cluster showing the largest loss-of-function phenotype (Fig. 3g and Supplementary Fig 3c).

## Neutralization of SINV-EEEV infection by LA3 tandem repeat domain proteins

Our BLI data suggested that LDLR interaction with EEEV VLPs was low affinity in nature since binding occurred when the receptor was in the solid phase but not in solution. Based on these results, we anticipated that in contrast to that seen with MXRA8-CHIKV[18,19] and LDLRAD3-VEEV[23], which had monovalent $K_D$ binding affinities of approximately 80 to 200 nM, a soluble LDLR decoy molecule might not inhibit SINV-EEEV infection. Indeed, pre-incubation of SINV-EEEV with LDLR-Fc or LBD-Fc did not block infection of LDLR-expressing THP-1 cells (Fig. 4a) or N2a *ΔB4galt7* cells (Supplementary Fig 4a) even at relatively high (10 μg/ml) concentrations, whereas they did block infection by VSV, an unrelated

rhabdovirus that can use LDLR as an entry receptor[39,40]. Similarly, pre-incubation with LA3-Fc did not inhibit SINV-EEEV infection but completely blocked VSV infection (Fig. 4b). We hypothesized that if we could increase the valency of the decoy molecule, we might enhance avidity enough to block infection. To test this idea, we designed LA3-Fc fusion proteins with three (LA3-LA3-LA3-Fc [LA3(3)-Fc]) or five (LA3-LA3-LA3-LA3-LA3-Fc [LA3(5)-Fc]) copies of the LA3 domain (Supplementary Fig 4b). Indeed, soluble LA3(3)-Fc and LA3(5)-Fc proteins bound to immobilized EEEV VLPs by ELISA, with LA3(5)-Fc showing an approximately 100-fold increase in binding avidity (Fig. 4c).

We next tested whether LA3(5)-Fc could inhibit infection of SINV-EEEV. Indeed, LA3(5)-Fc, but not LA3(3)-Fc, blocked SINV-EEEV infection, although efficient inhibition occurred only at relatively high concentrations (Fig. 4d). We increased the valency of LA3 further by engineering LA3(5) into the pentamerization domain of cartilage oligomeric matrix protein (COMP) using a 27-amino acid (Gly-Gly-Ser)₉ linker (Fig. 4e and Supplementary Fig 4c). The LA3(5)-COMP protein efficiently and dose-dependently blocked SINV-EEEV infection in human THP-1 cells and SH-SY5Y neuronal cells, whereas LDLRAD3-D1-COMP did not (Fig. 4f and Supplementary Fig 4d). We observed similar neutralization results with LA3(5)-COMP and SINV-WEEV (Fig. 4g). This result was confirmed by BLI analysis, which showed binding of LA3(5)-COMP to WEEV VLPs in both orientations, in the solid phase or in solution (Supplementary Fig 4e, f). In contrast, SINV-SFV infection was not inhibited by LA3(5)-COMP (Fig. 4h). Consistent with this data, while SFV VLPs bound the full length LDLR-Fc in the solid-phase, only minimal binding was detected to LA4-Fc and virtually none to LA3-Fc (Supplementary Fig 4g, h).

## In vivo protection by LA3 tandem repeat proteins against EEEV-MADV

To assess the physiological role of the interaction between LDLR and EEEV, we evaluated whether co-injection of LA3(5)-COMP with MADV (Argentina 1936) would protect C57BL/6J mice from infection. Notably, less weight loss and greater survival rates after MADV infection were

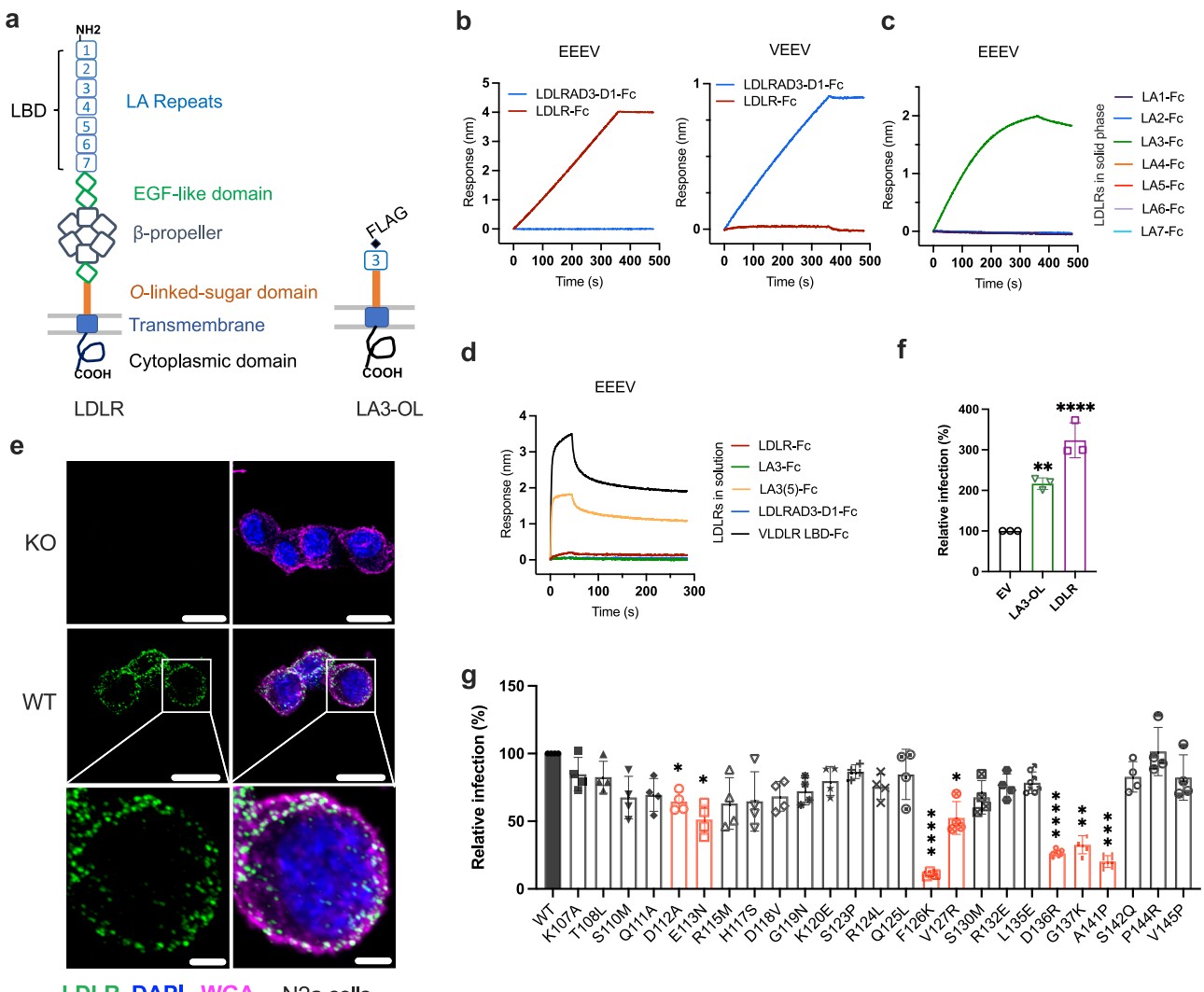

**Fig. 3 | LDLR directly binds to EEEV virions. a** Schematic of domain structure of LDLR (left) and LA3-OL mini receptor (right). **b–d** Biolayer interferometry (BLI) graphs showing EEEV VLPs binding to immobilized full length LDLR-Fc (LDLR-Fc) (**b**), EEEV VLPs binding to immobilized single LDLR LA domains (LA-Fc) (**c**), and LDLR proteins binding to immobilized EEEV VLPs (**d**) (one of two experiments is shown). **e** Confocal microscopy images of LDLR puncta stained on the plasma membrane surface of N2a cells. LDLR (green), nuclei (DAPI, blue), and cell membrane (wheat germ agglutinin, WGA, magenta). WT denotes *ΔB4galt7* N2a cells, and KO denotes *ΔB4galt7 ΔLdlr* N2a cells. Scale bar, 10 μm for low magnification images and 2 μm for zoomed in images (*n* = 2 experiments, with representative fields shown). **f** LA3 mini-receptor LA3-OL promotes infection of SINV-EEEV-GFP in

transduced THP-1 cells (*n* = 3 experiments). **g** Mapping of the LDLR LA3 domain binding site for EEEV. THP-1 cells ectopically expressing indicated mutations were inoculated with SINV-EEEV-GFP. Cell surface expression data of mutants is shown in Supplementary Fig 3e. Red columns indicate mutations that significantly impair the ability of the LA3 domain to support SINV-EEEV infection (*n* = 4 experiments). Statistical analysis (*P* values from left to right): one-way ANOVA with Dunnett's post-test: (**f**) **\*\*P* = 0.0028, \*\*\*\*P* < 0.0001; (**g**) one-way ANOVA with Welch and Brown-Forsythe post-test: \*P = 0.0185, \*P = 0.0252, \*\*\*\*P < 0.0001, \*P = 0.0336, \*\*\*\*P < 0.0001, \*\*P = 0.0022, \*\*\*P = 0.0003. Column heights indicate mean values, and error bars denote SD. Source data are provided as a Source Data file.

observed in mice that received LA3(5)-COMP compared to the LDLRAD3-D1-COMP control protein (Fig. 4i, j). After treatment, we also quantified MADV RNA levels at days 3 or 5 post-infection (dpi) in representative lymphoid (spleen), visceral (kidney), and central nervous system (brain) tissues. Less viral RNA was detected in LA3(5)-COMP treated than LDLRAD3-D1-COMP treated mice in the spleen at 3 dpi (112-fold, *P* < 0.01) and in kidney at 5 dpi (5-fold, *P* < 0.05) (Fig. 4k). Treatment with LA3(5)-COMP also resulted in lower levels of MADV RNA in the brain at 3 and 5 dpi (7 to 954-fold, *P* < 0.01) compared to mice administered LDLRAD3-D1-COMP (Fig. 4k). To corroborate these findings, we also measured infectious virus levels in brain by focus-forming assays. At 5 dpi, mice treated with LA3(5)-COMP had substantially less (1,200-fold, *P* < 0.01) infectious virus in the brain than animals administered LDLRAD3-D1-COMP (Fig. 4l).

Using an available Mouse Cell Atlas database (https://bis.zju.edu.cn/MCA/), we noted that *Ldlr* mRNA expression in the adult mouse brain extends to oligodendrocytes, astrocytes, microglia, but interestingly, not neurons (Supplementary Fig 5a, b). Based on these data, and our LA3(5)-COMP protection experiments, LDLR might be required for EEEV neuropathogenesis by facilitating infection in some, but not all, brain cell types.

## Discussion

In this study, using an unbiased CRISPR/Cas9 genome-wide screen in cells of neuronal origin, we identified LDLR as a candidate attachment and entry receptor for EEEV and showed that it also can be used by WEEV, and to a lesser extent by SFV and possibly GETV. We validated LDLR as a receptor for EEEV in cell culture using a series of gene

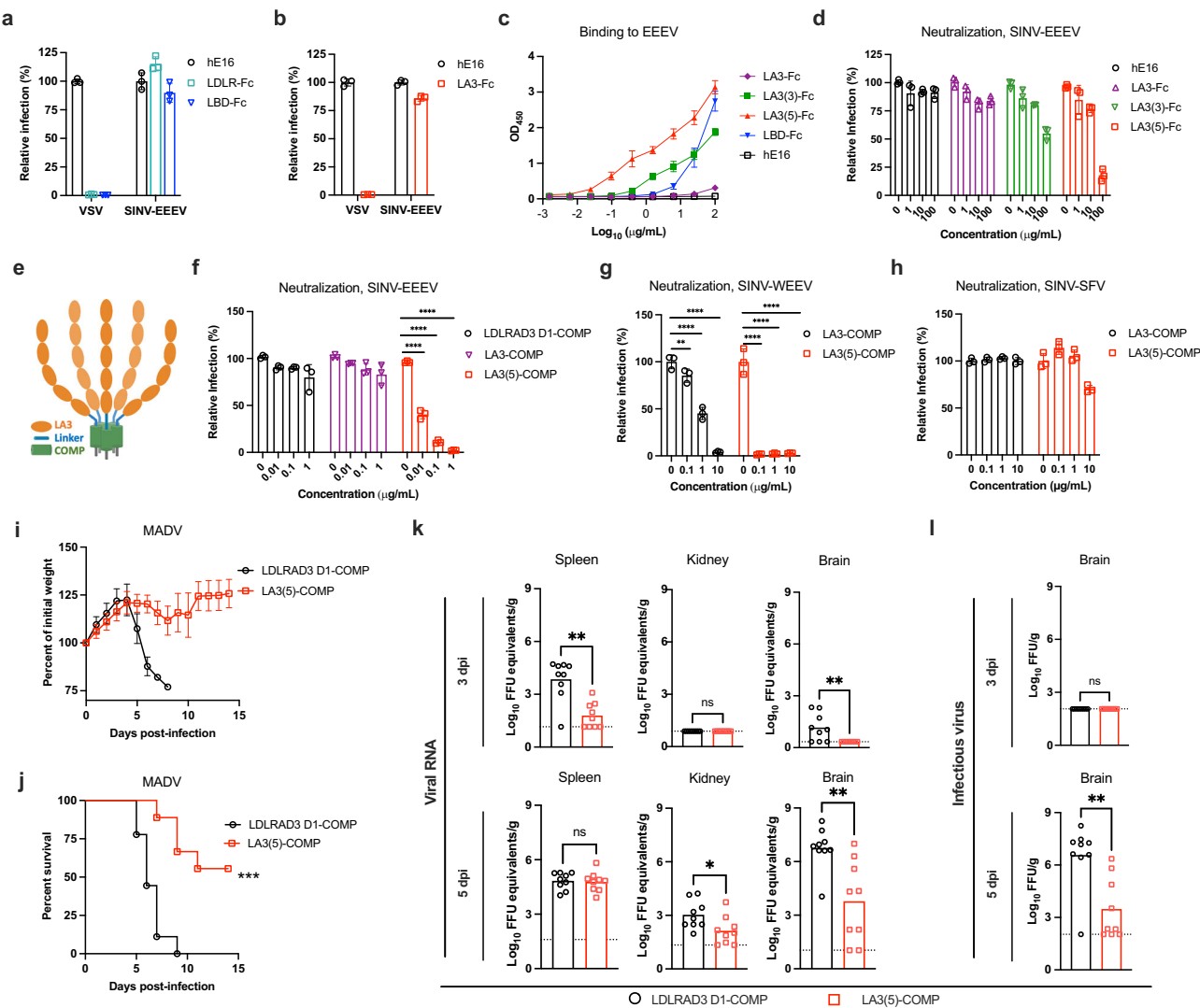

**Fig. 4 | Neutralization and protection of alphavirus infection by LA3 tandem repeat domain proteins. a, b** Effects of preincubation of LDLR-Fc (**a**), LBD-Fc (**a**), or LA3-Fc (**b**) on VSV and SINV-EEEV infection of THP-1 cells ectopically expressing LDLR (n = 3 experiments). hE16 is an anti-West Nile virus mAb (negative control). **c** Binding of soluble LA domain-Fc molecules with repeat domains to immobilized EEEV as measured by ELISA (n = 3 experiments). **d** Dose-dependent effect of pre-incubation of LA3-Fc, LA3(3)-Fc, and LA3(5)-Fc on SINV-EEEV infection (n = 3 experiments). **e**, Schematic of a pentamer of LA3(5). **f–h** Effect of pre-incubation of LA3(5)-COMP (or LDLRAD3 D1-COMP, negative control) protein on infection by SINV-EEEV (**f**), SINV-WEEV (**g**), or SINV-SFV (**h**) in THP-1 cells ectopically expressing LDLR (n = 3 experiments). **i–l** Four-week-old C57BL/6J mice were inoculated subcutaneously with 2 x 10² FFU of MADV (Argentina 1936) pre-mixed with 50 µg of LA3(5)-COMP or LDLRAD3-D1-COMP. A second 100 µg dose of LA3(5)-COMP or LDLRAD3-D1-COMP was administered by intraperitoneal injection at day +1. **i** Weights were measured daily through the duration of the experiment. **j** Survival was monitored through 14 days (**i**, **j**: n = 9, three experiments). **k, l** At 3 or 5 days post-infection, the indicated tissues were collected (n = 9 two experiments), and MADV RNA levels were quantified by RT-qPCR (**k**) and normalized to a standard curve from a stock of known titer. Infectious virus was quantified by focus-forming assay (**l**). Limits of detection (LoD) are indicated by dashes lines. Means ± SD are shown. Statistical analysis (P values from left to right): **f, g** one-way ANOVA with Dunnett's post-test: ****P < 0.0001 (**f**). ****P < 0.0001, **P = 0.0087(**g**); **j** Log-rank (Mantel-Cox) test, ***P = 0.0003; **k, l** two-tailed Mann–Whitney test: **P = 0.0026 (spleen, 3 dpi), **P = 0.0090 (brain, 3 dpi), *P = 0.0267 (kidney, 5 dpi), **P = 0.0019 (brain, 5 dpi) (**k**); **l** **P = 0.0019; ns, not significant. Source data are provided as a Source Data file.

editing, complementation, attachment and internalization, direct binding, and neutralization experiments. Moreover, domain binding and neutralization studies identified an interaction between EEEV and WEEV and the LA3 domain but not for SFV. Our study expands on the number of recently described LDLR family members that can act as entry receptors for different alphaviruses, including LDLRAD3 (VEEV only), VLDLR (EEEV, SFV, and possibly SINV), and ApoER2 (EEEV, SFV, and possibly SINV)[23,26]. LDLR appears to be a common target of viruses and has been described as an entry receptor for VSV[39], a minor group of rhinoviruses[41], and hepatitis C virus[42].

In contrast to other alphavirus-receptor interactions (*e.g.*, VEEV-LDLRAD3[23,25] and CHIKV-MXRA8[18–20]), the affinity of LDLR for EEEV appears low. Whereas EEEV virions could bind readily to immobilized

LDLR or specific LA domains in an ELISA or BLI format, soluble LDLR or LA domains showed poor, if any, binding to immobilized virions. EEEV, WEEV, and possibly other alphaviruses can engage LDLR because of multivalent binding to clustered LDLR molecules on the cell surface, which yields an avidity benefit; the avidity of binding may be further enhanced by the number of E2-E1 binding sites on the icosahedral alphavirus virion. While further studies are warranted, low-affinity binding receptors could have an advantage in alphavirus infection by facilitating the release of virions for endosomal fusion and enabling penetration in the cytosol.

In previous studies, we and others generated soluble alphavirus receptor decoy molecules by fusing single, two, or multiple domains to an Fc moiety and demonstrated their neutralizing and protective

potential in vitro and in vivo[18,23,26]. The low-affinity nature of the interaction between EEEV and LDLR posed a challenge for this, since high (100 µg/ml) concentrations of LDLR-Fc or LA3-Fc showed little inhibitory activity in cell culture. To overcome this limitation, we designed LA3 domain proteins with 3 or 5 tandem repeats and expressed them in the context of an Fc-fusion protein (valency of 6 or 10) or a COMP pentamerized protein (valency of 25). These proteins showed markedly greater inhibitory activity against EEEV and WEEV in cell culture and protected mice against MADV infection in several tissues including the brain as well as limiting weight loss and mortality. Thus, these multimerized decoy receptors serve as a proof-of-principle for LDLR interactions with EEEV or EEEV-lineage virions in vitro and in vivo.

LDLR family proteins are defined by their conserved LA domain repeats. Different viruses appear to have distinct preferences for binding of LA domains: VEEV exclusively uses the LA1 domain of LDLRAD3[23–25], SFV preferentially uses the LA3 domain of VLDLR[27], and VSV uses LA2 or LA3 domains of LDLR[40]. Our scanning mutagenesis mapping studies identified a conserved aromatic phenylalanine (Phe126) that was essential for LA3-mediated binding and EEEV infection. When Phe126 was mutated to a Lys residue, which is present in the nonbinding LA7 domain, the LA3 domain no longer supported EEEV infection. This same Phe126 residue is critical for the interaction of VSV with LDLR LA3[40], and analogous aromatic residues contribute to the binding of VSV with LDLR LA2 (Trp97)[40], VEEV with LDLRAD3 (Trp47)[24,25], SFV with VLDLR (Trp132)[27], and human rhinovirus 2 (HRV2) with VLDLR (Trp132)[43], suggesting EEEV may recognize LDLR through a similar mode of engagement. We also identified a key alanine residue (Ala141) as important for LDLR LA3 interaction with EEEV. Intriguingly, this amino acid is not found in other LA domains in LDLR and could contribute to the LA domain preference for EEEV. Structural studies are planned to directly address these hypotheses.

In summary, our CRISPR/Cas9 screen identified LDLR as a low affinity, yet potentially high avidity entry receptor for EEEV, WEEV, and possibly SFV. Given the increasing number and complexity of receptors for these alphaviruses, pathogenesis studies in knockout (KO) mice may not be as revealing as for those alphaviruses with dominant receptors (e.g., MXRA8 KO and CHIKV[17] or LDLRAD3 KO and VEEV[23]) unless a key cell type (*e.g.*, neurons, astrocytes, or microglia) uses a single receptor for entry or the individual receptors are expressed on different cell types. Notwithstanding, the development of soluble LDL family receptor decoys may provide a unique approach toward developing countermeasures against multiple virus family members.

# Methods
## Cells and viruses
Neuro 2a (N2a, ATCC CCL-131) and HEK 293T (ATCC CRL-3216) cells were cultured at 37°C in Dulbecco's modified Eagle medium (DMEM) with 10% fetal bovine serum (FBS), 10 mM HEPES, 100 U/mL penicillin and 100 U/mL streptomycin. THP-1 cells (ATCC TIB-202) were cultured at 37°C in RPMI with 10% fetal bovine serum (FBS), 10 mM HEPES, 100 U/mL penicillin and 100 U/mL streptomycin. For genetically modified N2a cells, selection was maintained using the following antibiotics: puromycin (2.5 µg/mL, InvivoGen), blasticidin (4 µg/mL, InvivoGen) or hygromycin (200 µg/mL, InvivoGen). Cell viability and cytotoxicity assays were conducted using CellTiter-Glo (Promega) as per the manufacturer's instructions. Luminescence was read on a Synergy H1 Hybrid Multi-Mode Reader (BioTek) at room temperature with 0.5 second integration per well. All cell lines were tested routinely for mycoplasma infection and found negative. All requests for resources and reagents should be directed to the Corresponding author. This includes viruses, proteins, and cells. All reagents will be made available on request after completion of a Materials Transfer Agreement (MTA)

The following viruses were used: SINV-EEEV (FL93-939), SINV-VEEV (TrD), SINV-WEEV (CBA87), SINV-CHIKV (LR 2006), SINV

(TR339), SINV-SFV (SFV4), MAYV (BeH407), GETV (AMM-2021), VSV (Indiana), and MADV (Peru 1970, Argentina 1936, Brazil 1975, Brazil 1985, and Colombia 1992. Supplementary Table 1). All MADV strains were obtained from the World Reference Center for Emerging Viruses and Arboviruses (WRCEVA, Galveston, TC, kind gift of S. Weaver and K. Plante). Replication-competent chimeric SINV viruses expressing GFP were generated by replacing the SINV TR339 structural proteins with VEEV, EEEV, WEEV, SFV, or CHIKV structural proteins[44,45,46]. Viruses were propagated in BHK-21 cells or C6/36 cells and titrated by focus-forming or plaque assay in Vero cells.

## CRISPR–Cas9 screen and data analysis
The CRISPR screen was performed using a clonal glycosaminoglycan-deficient *ΔB4galt7* N2a cell line[23]. The genome-wide CRISPR–Cas9 screen was performed using the mouse GeCKO v.2 CRISPR knockout pooled library[28] (Addgene, 1000000053) containing 130,209 sgRNAs targeting 20,611 genes. A *ΔB4galt7* N2a cell with GeCKO v.2 CRISPR knockout pooled library was made as described previously[23]. For each half-library, $1 \times 10^8$ sgRNA-containing cells were seeded into ten 175-cm$^2$ tissue culture flasks, cultured for 20 h, and inoculated with SINV-EEEV (FL93-939) at an MOI of 5. Four days after inoculation, surviving cells were collected and cultured in DMEM supplemented with 10% FBS and a cocktail (2 µg/mL) of anti-EEEV neutralizing mAbs (EEEV-3, EEEV-10 and EEEV-102[45]). Expanded cells were re-inoculated with SINV-EEEV for two additional rounds of screening. Genomic DNA was extracted from uninfected control cells ($3 \times 10^7$ per sub library) and surviving cells ($1 \times 10^7$ per repeat). The sgRNAs were enriched, amplified, and sequenced using an Illumina HiSeq 2500 (Genome Technology Access Center of Washington University). The sgRNA sequences against specific genes were obtained after the removal of the tag sequences using the FASTX-Toolkit (http://hannonlab.cshl.edu/fastx_toolkit/) and cutadapt (version 1.8.1). sgRNA sequences were analyzed using a published computational tool (MAGeCK)[47] (Supplementary Data 1).

## Gene validation
Five candidate genes were validated using two or three sgRNAs to the genes. A sgRNA that does not target the mouse or human genomes was included as a negative control. The sequences of the guides are listed in Supplementary Table 2. The sgRNAs were cloned into lentiCRISPR v.2 (Addgene, 52961) and packaged in HEK 293T cells with psPAX2 (Addgene, 12260) and pMD2.G (Addgene, 2259) using Lipofectamine 3000 (Thermo Fisher). *ΔB4galt7* N2a cells[23] were transduced and selected for 7 days in the presence of puromycin. Clonal LDLR-deficient cells were obtained by limiting dilution. *Ldlr* gene editing was validated by next-generation sequencing on an Illumina HiSeq 2500 platform (Genome Technology Access Center of Washington University) with 300-base-pair paired-end sequencing. *Ldlr* gene editing also was confirmed by flow cytometry analysis of LDLR surface expression using a goat-anti-mouse LDLR antibody (Thermo Fisher, PA5-46987, 1:100 dilution) and detection with a donkey-anti-goat-AF488 (Thermo Fisher, A-11055, at dilution of 1:1,000). When staining THP-1 cells, a 1:200 dilution of Fc block (Biolegend, 422301) was added according to the manufacturer's instructions.

## Complementation and ectopic expression of LDLR
The mouse *Ldlr* gene (ENSMUST00000034713.9; https://useast.ensembl.org/Mus_musculus/Transcript/Summary?db=core;g=ENSMUSG00000032193;r=9:21634779-21661215;t=ENSMUST00000034713] and human *LDLR* gene (ENST00000558518.6; https://useast.ensembl.org/Homo_sapiens/Transcript/Summary?db=core;g=ENSG00000130164;r=19:11089463-11133820;t=ENST00000558518]) were synthesized, and cloned into the lentivirus vector pLV-EF1a-IRES-Hygro (Addgene, 85134) between the BamHI and MluI restriction enzyme sites (Azenta Life Sciences). The *Ldlr* sgRNA target sequence was mutated in the

seed sequence and subsequent PAM sequence synonymously (from TGACCGTGAACATGACTGCAAG to TGACCGTGAACATGAtTGtAAa) to prevent editing of the re-introduced gene. An N-terminal Flag tag downstream of the signal sequence was added to monitor cell surface expression. The LDLR-encoding vectors were packaged as lentiviruses, and cells were transduced as described above. Complemented cells were selected with hygromycin (200 µg/mL) for at least 5 days before use. Complemented cells were assessed for LDLR surface expression using an anti-Flag antibody (Cell Signaling Technology, 14793) and Alexa Fluor 647-conjugated goat anti-rabbit IgG (Thermo Fisher, A27040) or anti-LDLR antibody (Thermo Fisher, PA5-46987, 1:200) and Alexa Fluor 647-conjugated secondary antibody (Thermo Fisher, A-11055, 1:1000). Stained cells were analyzed on a MACSQuant Analyzer 10 (Miltenyi Biotec). The above-mentioned mouse *Ldlr* gene and human *LDLR* full length isoform were introduced into human THP-1 cells and selected with hygromycin (200 µg/mL) for two weeks before use.

### LA3 domain mutagenesis

A mutational library was generated by substituting single amino acid changes in the LA3 domain of LDLR protein. Amino acids essential for maintaining the three-dimensional structure of the protein (*i.e.*, cysteines that form disulfide bond, the amino acids that coordinate the calcium ion, and those forming the hydrophobic core) were not changed[48]. Other amino acids in LA3 domain were mutated according to BLOSUM scoring matrix[38]. The mutants were synthesized and cloned into the lentivirus vector pLV-EF1a-IRES-Hygro (Addgene, 85134) between the BamHI and MluI restriction enzyme sites (Genescript LLC). An N-terminal FLAG tag was added to each LA3 mutant to monitor expression. THP-1 cells were transduced to express each individual LA3 mutant and then inoculated with SINV-EEEV expressing GFP at an MOI of 10 for 20 h. Cells were also stained with an anti-FLAG antibody (D6W5B, Cell Signaling, USA) to measure the expression level of these mutants on cell surface.

### Infection assays

For single- (< 10 h) or multi- (< 24 h) step infection assays, *ΔB4galt7*, *ΔB4galt7 ΔLdlr*, and *Ldlr*-complemented N2a cells were inoculated with the following viruses: SINV-EEEV (FL93-939) (MOI of 100, 8.5 h or MOI of 10, 24 h), MADV (Peru 1970, MOI of 10, 8 h; Argentina 1936, MOI of 10, 8 h, Brazil 1975, MOI of 20, 8 h; Brazil 1985 MOI of 10, 8 h, Colombia 1992, MOI pf 20, 8 h), SINV-VEEV (TrD) (MOI of 20, 7.5 h), SINV-WEEV (CBA87) (MOI of 10, 18.5 h), SINV (TR339) (MOI of 1, 24 h), SINV-CHIKV (MOI of 10, 7.5 h), SINV-SFV (MOI of 0.3, 16 h), MAYV (MOI of 3, 18 h), and GETV (MOI of 1, 18 h). LDLR-expressing THP-1 cells were inoculated with SINV-EEEV (MOI of 10, 16 h), SINV-WEEV (MOI of 10, 24 h), or SINV-SFV (MOI of 10, 24 h), respectively. At indicated time points, cells were collected using trypsin and fixed with 1% or 2% paraformaldehyde (PFA) in PBS for 15 min at room temperature. Cells inoculated with GFP-containing viruses were analyzed on a MACSQuant Analyzer 10 (Miltenyi Biotec) or an iQue3 Cytometer (Sartorius). Cells infected with MADV, MAYV, or GETV were stained with DC2.112 mAb (1:1,000 dilution)[46], followed by goat-anti-human IgG (Thermo Fisher, A21445, 1:1,000). All flow cytometry data were processed using FlowJo (FlowJo 10.0).

For multi-step growth curve, virus (Argentina 1936 strain for MADV) was inoculated at MOI 0.01. Supernatant were sampled every 12 h for 60 h. Virus titer in the samples was determined with focus forming assay. THP-1 cells ectopically expressing human LDLR or LA domains were inoculated with SINV-EEEV (MOI of 10, 20 h) and analyzed with flow cytometry as described above.

### Virus binding and internalization assays

For virus-binding assays with THP-1 cells, SINV-EEEV virions (MOI of 1) were added to $10^5$ THP-1 cells in a 96-well plate and incubated on ice for 90 min. To remove unbound virions, cells were washed 6 times with ice-cold RPMI medium. Cells were collected, lysed in RLT buffer (Qiagen), and RNA extraction was performed using an RNeasy Mini Kit (Qiagen). For internalization assays, SINV-EEEV (MOI of 1) was added to $10^5$ cells in a 96-well plate and incubated on ice for 90 min. After 6 washes with ice-cold RPMI medium, pre-warmed 37°C medium supplemented with 10% FBS and 15 mM $NH_4Cl$ was added to cells. Cells were incubated at 37°C for 1 h to allow virus internalization. Cells were then chilled on ice and incubated with 0.5 mg/mL of proteinase K (Sigma, P2308) in PBS on ice for 2 h to digest and remove residual plasma-membrane-bound, uninternalized virions. After 6 additional washes with ice-cold plain RPMI media, cells were lysed in RLT buffer, and RNA was extracted. The RT–qPCR was performed using a TaqMan RNA-to-CT 1-Step Kit (Thermo Fisher) with *GAPDH* as an internal control. Primers and probes (all from IDT) used are as follows: SINV-EEEV forward: 5′-AAGATCATCGACGCAGTCATC-3′; SINV-EEEV reverse: 5′-GCTGTGGAAGTAACCGAATCT-3′; SINV-EEEV probe: 5′-/56-FAM/CCACCTTAC/ZEN/TTCTGCGGCGGATTTA/3IABkFQ/-3′. GAPDH forward: 5′-GCCCAGAACATCATCCCTGC-3′; GAPDH reverse: 5′-CCGTT CAGCTCTGGGATGACC-3′; and GAPDH probe: 5′ 6-FAM/ TCCACTGGT/ZEN/GCTGCCAAGGCTGTG/3′ IABkFQ.

For virus-binding assays with N2a cells, *ΔB4galt7 ΔLdlr* (*ΔLdlr* + EV), and LDLR-complemented *ΔB4galt7 ΔLdlr* (*ΔLdlr* + *Ldlr*) N2a cells were removed from plates by gentle scraping and resuspended in DMEM with 2% FBS. Cells were incubated with SINV-EEEV (MOI of 1) at 4°C for 45 min followed by six washes with DMEM with 2% FBS to remove unbound virus. Cells were lysed in RLT buffer, and RNA was extracted for RT-qPCR analysis. For the internalization assay, after binding and washing, cells were incubated at 37°C for 45 min. To remove uninternalized virus, cells were washed with DMEM twice and the incubated in DMEM supplemented with 100 µg/mL of proteinase K (Thermo Scientific, EO0491) at 37°C for 15 min. After two additional washes with DMEM with 2% FBS, cells were incubated with 100 µg/mL of RNAse A (Thermo Scientific, EN0531) in DMEM with 2% FBS at 37°C for 30 min. After two final washes with DMEM with 2% FBS, cells were lysed in RLT buffer, and RNA was extracted for RT-qPCR analysis.

### Recombinant LDLR protein generation and purification

To generate Fc fusion proteins, LDLR gene fragments (LBD, residues 25-313; LA1, residues 25-65; LA2, residues 66-106; LA3, residues 107-145; LA4, residues 146-186; LA5 residues 195-233; LA6, residues 234-272; LA7 residues 274-313; 3 copies of LA3 in tandem; 5 copies of LA3 in tandem) were codon-optimized, synthesized (Twist Bioscience), and inserted into the pTwist vector with the mouse IgG heavy chain signal peptide sequence MGWSCIILFLVATATGVHS, a linker of SSSGSSG, and the human IgG1 Fc region Asp 104 – Lys 330 (P01857.1) with L234A, L235A, and P329G mutations to abolish binding to cell surface Fc-gamma receptors. A Fc fusion protein of LDLRAD3 domain1 (D1) was constructed as described previously[23]. HRV 3C cleavable constructs contained the cleavage site (LEVLFQGP) immediately downstream of the LDLR coding sequence. The residue number follows Uniprot P01130.1. A commercial LDLR-Fc protein (LDR-H5254, Acrobiosystems) containing the ectodomain (Ala 22 – Arg 788) fused with human IgG1 Fc region (Pro 100 – Lys 330, P01857.1) was included as the full-length LDLR.

To create proteins fused with the pentamerization domain of cartilage oligomeric matrix protein (COMP)[49], the LA3 domain of LDLR designed above (three or five tandem repeats) or LA1 domain of LDLRAD3 was cloned ahead of a GGS linker of 27 residues, the human COMP pentamerization domain (MDLAPQMLRELQETNAALQDV-RELLRQQVKEITFLKNTVMECDAC), a GGGSGGS linker of 27 residues and a Strep-tag WSHPQFEK for purification. Constructs were confirmed by Sanger sequencing.

Human HEK293 cells were co-transfected with the constructs of Fc- or COMP-fusion proteins and the human receptor-associated

protein RAP (*LRPAP1*, NM_002337.4), a chaperone that enhances cell-surface expression of LDLR family members[50]. To further enhance the expression levels, we modified *LRPAP1* gene by deleting the ER retention signal (HNEL) at the C-terminus.

Expi293 cells (200 mL) were seeded at $3 \times 10^6$ cells per mL on the day of transfection. Fc or COMP fusion protein constructs (130 µg) and RAP (70 µg) plasmids were diluted in Opti-MEM, complexed with ExpiFectamine 293 transfection reagent (Thermo Fisher) and added to cells. One day later, cells were supplemented with ExpiFectamine 293 transfection enhancers # 1 and 2 to boost transfection efficiency. Supernatant was harvested 4 days after transfection, centrifuged at $3,000 \times g$ for 20 min, and purified using protein G sepharose 4B (Thermo Fisher) for Fc fusion proteins or IBA Strep-Tactin XT 4Flow resin (IBA Life sciences) for COMP fusion proteins as per the manufacturer's instructions. An additional washing step was added before elution with wash buffer supplemented 50 mM EDTA to remove the RAP protein bound to LDLR LA domains. After elution, the eluate was concentrated with a spin concentrator with cut off value of 10 kDa (Millipore). The buffer was changed to 1 x Tris buffered saline (TBS) using Zeba desalting column (Thermo Fisher). $CaCl_2$, final 2 mM concentration, was added to the proteins in 1x TBS buffer. Protein purity was confirmed by SDS–PAGE.

## ELISA

Nunc MaxiSorp ELISA plates (Thermo Fisher) were coated with LDLR Fc-fusion proteins at 4°C overnight in sodium bicarbonate buffer, pH 9.3. Positive (anti-alphavirus DC2.112[46]) and negative (anti-West Nile Virus antibody hE16[51]) control were included when needed. Plates were washed with PBS/Tween and blocked with PBS containing 2%(w/v) BSA for 1 h at room temperature. EEEV VLPs (1 µg/mL) were diluted in PBS containing 2% BSA and added to wells for 1.5 h at room temperature. Plates were washed with 1x PBS / 0.05% Tween-20 and incubated with a mouse anti-EEEV antibody, EEEV-10[45] at 1 µg/mL at room temperature for 1.5 h. Plates were washed with PBS/Tween and detected with anti-mouse IgG conjugated with horseradish peroxidase (Sigma, A8924, 1:10,000) in a blocking buffer at room temperature for 30 min. Plates were washed with PBS/Tween and developed using 3,3′-5,5′-tetramethylbenzidine substrate (Thermo Fisher, 34028) at room temperature for 10 to 15 min, stopped with 2N $H_2SO_4$, and read at 450 nM using a TriStar Microplate Reader (Berthold).

To test LDLR or its LA domains for binding to EEEV VLPs, plates were coated with anti-EEEV mAb EEEV-10 at 1 µg/mL overnight. Plates were washed with PBS and blocked with PBS containing 2% (w/v) BSA for 1 h at room temperature. EEEV VLPs (1 µg/mL) were diluted in PBS containing 2% BSA and added to wells for 1 h at room temperature. Fc-fused LDLR proteins or domains were diluted in 2% BSA and incubated for 1 h at room temperature. Positive (anti-alphavirus DC2.112) and negative (anti-West Nile virus (WNV) antibody hE16[51]) control were included when needed. Plates were washed with PBS and incubated with horseradish-peroxide-conjugated goat anti-human IgG (H + L) (Sigma, A6029) for 1 h at room temperature. Plates were developed as described above.

## LDLR protein neutralization assays

SINV-EEEV expressing GFP (MOI of 10) or VSV expressing GFP (MOI of 1) were pre-incubated at 37°C with LDLR–Fc, LBD-Fc, LA3-Fc, LA4-Fc, LA5-Fc, LA3(3)-Fc, LA3(5)-Fc at 10 µg/mL or 0 to 100 µg/mL with 10-fold serial dilutions in DMEM or RPMI supplemented with 2% FBS. Anti-WNV mAb E16 was included as a negative isotype control. The complexes were added to either *ΔB4galt7* N2a cells or LDLR-expressing THP-1 cells for 18 h. Cells were analyzed for GFP expression by flow cytometry using a MACSQuant Analyzer 10 (Miltenyi Biotec) or iQue3 (Satorius). For neutralization experiments with LA3(5)-COMP, proteins were diluted serially in DMEM supplemented with 2% FBS with final concentrations ranging from 0 to 1 or 0 to 10 µg/mL. SINV-EEEV, SINV-WEEV, or SINV-SFV (all at MOI of 10) were pre-incubated with proteins at 37°C for 45 min, and the mixture was then added to LDLR-expressing THP-1 cells for 18 h or SH-SY5Y neuronal cells for 8 h (SINV-EEEV), 20 h (SINV-WEEV), or 24 h (SINV-SFV).

## Immunofluorescence microscopy

For cell surface staining of LDLR, $3 \times 10^4$ *ΔB4galt7* (WT) or *ΔB4galt7 ΔLdlr* N2a cells (KO) were seeded into wells of a chamber slide (Nunc, Lab-Tek II) in 300 µL of DMEM media and cultured for 24 h. Cells were washed with pre-warmed DMEM, incubated in 250 µL of DMEM at 37°C for 30 min, washed with cold wash buffer (HBSS, 20 mM HEPES, and 1% BSA), blocked with cold blocking buffer (5% normal donkey serum in wash buffer) on ice for 30 min, and then stained with 150 µL of anti-LDLR (Thermo Fisher, PA5-46987,1:50) in 2.5% normal donkey serum in wash buffer on ice for 30 min. After three additional 5-min washes with cold wash buffer, cells were stained with donkey-anti-goat IgG-AF488 (Thermo Fisher, A-11055,1:500 in staining buffer) on ice for 30 min in the dark. Cells were washed with cold washing buffer three times with cold HBSS buffer supplemented with 20 mM HEPES before fixing in 4% formaldehyde in HBSS for 10 min at room temperature. Cells were washed three additional times with HBSS buffer, stained with WGA-A633 (1:200 in PBS), and nuclei were counterstained with Hoechst 33258 (Invitrogen, H3569,1:5,000) for 15 min at room temperature. Slides were mounted with mounting medium (Invitrogen, F36980) and covered with cover glass.

For LDLR-expressing THP-1 cells, chamber slides were coated with 20 µg/mL of poly-D-lysine in PBS (with calcium and magnesium, Sigma D8662) at 37°C for 30 min. 3 x 10^4 cells were seeded in 300 µL of PBS per well and incubated for 15 min at 4 °C. Cells were washed twice with cold wash buffer, blocked with cold blocking buffer supplemented with 1:200 Fc receptor blocking solution (FcX, Biolegend 422302) on ice for 30 min and the staining, fixation and mounting were performed as described above for N2a cells.

Confocal microscopy was performed on a Zeiss LSM 880 confocal microscope using a Plan-Apochromat 63X (NA 1.4) DIC objective. Images in each experiment were acquired at identical settings. Images were processed and analyzed with ImageJ software v.1.53t.

## Biolayer interferometry

Experiments were performed on a GatorPlus BLI (GatorBio) with all samples prepared in PBS supplemented with 1% BSA and 2 mM $CaCl_2$ (running buffer). LDLR Fc fusion constructs were captured on anti-human IgG Fc biosensors (GatorBio #160003) for 120 s, washed in running buffer for 30 s, then submerged into VLPs at a nominal concentration of ~10 µg/mL for 360 s. To assess the interaction between VLPs and receptor constructs in solution, a murine mAb (EEEV-3[45], WEEV-209, or CHK-124[52]) was immobilized on anti-mouse IgG Fc biosensors (GatorBio #160004) then used to capture EEEV, WEEV or SFV VLPs at a nominal concentration of ~10 µg/mL for 240 s. After washing in running buffer for 90 s, VLP-coated biosensors then were dipped into wells containing 1 µM of the designated Fc-fusion construct. Raw BLI response traces were processed using GatorOne Software v2.7 (GatorBio).

## Mouse experiments

C57BL/6J mice were purchased from Jackson Laboratories (Cat #000664) and maintained in a specific-pathogen-free facility. Four-week-old male mice were used in all experiments. Experimental procedures were approved by the Washington University School of Medicine Institutional Animal Care and Use Committee (assurance number A3381-01) and followed guidelines of the *Guide for the Care and Use of Laboratory Animals*. Virus inoculations were performed under anesthesia induced by ketamine hydrochloride and xylazine.

C57BL/6J mice were inoculated subcutaneously in the left footpad with 2 x 10^2 FFU of MADV (Argentina 1936) that was pre-incubated with

50 µg of LA3x5-COMP or D1-COMP for 45 min at 37 °C. At 24 hours post-infection, mice were administered an additional 100 µg dose of LA3x5-COMP or D1-COMP via intraperitoneal injection. Mice were monitored daily for weight loss and survival. To quantify viral RNA, at 3 or 5 dpi, mice were perfused, and spleens, kidneys, brains, and serum were collected, weighed, and homogenized using a bead homogenizer. RNA was extracted from supernatants of homogenized and clarified tissues using MagMax-96 Viral RNA Isolation kits (ThermoFisher) and a KingFisher Flex system (ThermoFisher) as specified by the manufacturer. Viral RNA was quantified by RT-qPCR using a TaqMan RNA-to-$C_T$ 1-Step kit (ThermoFisher). FFU equivalents were determined using a standard curve generated by serial dilutions of RNA extracted from a viral stock of known titer. The following primers were used to quantify virus RNA: MADV nsP3 forward: 5′-ATGAAGCGGGTGCGTATATC-3′, MADV nsP3 reverse: 5′-CAGAATAGGGTGCTGGAGTTT-3′, MADV probe 5′-/56-FAM/ATTTCTGTT/ZEN/GCAGATGCCCTTGCC/3IABkFQ/-3′.

To quantify infectious virus, the supernatants of clarified tissue homogenates were serially diluted ten-fold in DMEM with 2% serum. Vero cells were seeded at 3 x 10$^4$ cells per well in a 96-well flat-bottom plate and incubated overnight at 37°C in 5% $CO_2$. Then, 100 µL of the diluted tissue homogenates were added to Vero cells and incubated at 37°C in 5% $CO_2$ for 1 h. Subsequently, 100 µL of 1% methylcellulose in Minimum Essential Medium (MEM; Sigma #M0275) was overlaid on the cells, which were then incubated at 37°C in 5% $CO_2$ for 20 h. Then, the methylcellulose overlay was removed, and cells were fixed in 200 µL of 4% paraformaldehyde in PBS for 1 h at room temperature. Cells were rinsed with PBS and stained with 1 µg/mL of anti-alphavirus E1 antibody (DC2.112[46]) overnight at 4°C. The primary antibody was then rinsed off using PBS supplemented with 0.05% Tween-20, and samples were incubated in 1 µg/mL of horseradish peroxidase-conjugated goat anti-human secondary antibody (Sigma #A6029) at room temperature for 1 h. After rinsing, foci were developed using TrueBlue Peroxidase Substrate (KPL) per manufacturer's instructions and counted using a CTL-S6 Universal Analyzer (ImmunoSpot).

### Statistical analysis

Statistical significance was assigned when *P* values were < 0.05 using Prism (Version 8, GraphPad) and is indicated in each of the Figure legends. Cell culture experiments were analyzed by student's t-test, or one-way and two-way ANOVA with a multiple comparison correction. In animal experiments, survival differences were determined by a log-rank test, and virological data was analyzed using a Mann-Whitney test.

### Reporting summary

Further information on research design is available in the Nature Portfolio Reporting Summary linked to this article.

## Data availability

All data supporting the findings of this study are available within the main text and supplemental data. Source data for main and supplemental figures are provided with this paper. All reagents will be made available on request after completion of an MTA. Source data are provided with this paper.

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

## Acknowledgements

We acknowledge K. Carlton and J. Mascola and the Vaccine Research Center of the National Institutes of Allergy and Infectious Diseases (NIH) for a gift of the EEEV and WEEV VLPs. The studies were supported by NIH grants R01 AI141436 (M.S.D.) and R01AI095436 (W.B.K.), contract AI201800001 (to M.S.D.), U19 AI142790 (to M.S.D and D.H.F.), F30AI164842 (to N.M.K.), and Defense Threat Reduction Agency grant MCDC2103-11 (to W.B.K. and M.S.D.).

## Author contributions

H.M. and M.S.D. designed the study. H.M. performed the CRISPR–Cas9 screening, gene validation, infection studies, and mutagenesis experiments. H.M., S.R., and L.J.A. generated recombinant proteins and performed binding studies. A.S. and N.M.K. performed infection experiments with MADV. H.J. took and processed confocal microscopy images. D.H.F. supervised the protein binding studies. D.H.F., W.B.K., and M.S.D. secured funding and resources for the studies. H.M. and M.S.D. wrote the initial draft with all other authors providing comments.

## Competing interests

M.S.D. is a consultant on the Scientific Advisory Board for Inbios, Ocugen, Vir Biotechnology, Topspin Therapeutics, and Moderna. The Diamond Laboratory has received unrelated funding support in sponsored research agreements from Emergent BioSolutions, Moderna, and Vir Biotechnology. D.H.F. is a founder of Courier Therapeutics and has received unrelated funding support from Emergent BioSolutions and Mallinckrodt Pharmaceuticals. All other authors declare no competing interests.
