## [Peer Review File · Nature Communications]

The low-density lipoprotein receptor promotes infection of multiple encephalitic alphavirusesREVIEWER COMMENTS

Reviewer #1 (Remarks to the Author):

This manuscript uses a variety of sophisticated approaches to report the identification of a new cellular entry receptor (LDLR) for specific members of the alphavirus family. Alphaviruses include many medically important pathogens, for which there are no licensed anti-virals or medicines. Overall the studies are multifaceted and well done. The authors make use of a comprehensive CRISPR-based screen in the neuroblastoma cell line N2a to identify LDLR and confirms its role through a variety of approaches. Viral attachment and internalisation studies were then performed using the monocytic cell line THP-1, which was engineered to over express LDLR. Furthermore, LDLR is shown convincingly to bind directly to VEEV using biolayer interferometry, and the precise binding site more specifically mapped to specific amino acids in the LA3 domain. Despite the low affinity nature of LA3 interactions with virus, the authors define a decoy composed of tandem LA3 repeat domains that limits virus infection of THP-1 cells over expressing LDLR. Their in vivo data is limited but preliminarily suggests a role for LA3-VEEV interactions. Together, these results, if further validated using the suggested changes below, should constitute an important advance in our understanding of alphavirus biology.

Major changes

Figure 2 and 4. Considering that the majority of data in figure 1 was established using N2a cells that express LDLR without experimental manipulation, it is not clear why studies in this figure 2 and 4 were not also performed in N2a cells +/- LDLR. These should ideally be done and compared, especially as LDLR was over expressed in THP-1 cells and may not be particularly representative. This is also the case for Figure 4, in which LD3 tandem repeats were used to suppress infection of THP-1 cells. Data generated using a cell type with endogenous LDLR expression are required.

The majority of studies using cells with endogenous LDLR expression were undertaken using N2a cells, a neuroblastoma cell line that is most similar to immature neuronal precursors found in vivo. It is not clear how relevant these findings are for their in vivo non-immortalised neuronal counterparts, which often lack expression of genes detected in less differentiated cell lines. The maturity status of neurons has a defining impact of cell susceptibility to alphavirus infection (e.g. see extensive studies on neuron maturity and SFV/SINV infection). The relative immaturity of N2a cells may make them poor representatives of most neuron cell types found in the developed brain. In the absence of CNS specific in vivo data, the authors should define expression of LDLR in different neurons and discuss these limitations.

Figure 4i,j. The in vivo data is limited and incomplete. It is not clear whether the in vivo phenotype induced by LA3(5)-COMP administration is CNS specific and e.g. due to interference specifically with neuronal infection by virus. A better description of the clinical phenotypes of mice succumbing (or reaching clinically defined end points) is necessary, as is the determination of virus dissemination to target tissues (e.g. lymphoid tissue and brain), with or without LA3(5)-COMP administration.

Minor changes:

Lines 42-43. It would be better to additionally define SFV as a member of the old world alphaviruses, especially as it is referred to later in the introduction and is one of the viruses studied here.

Lines 92-96. Clarify how substituting non-structural genes from EEEV (BSL3) with those from SINV (BSL2) makes the chimera suitable for work at BSL2. Is the assumption that virulence of EEEV depends on non-structural gene expression? Please clarify and reference as appropriate.

Figure 2. All timing information of these assays is missing from legend and main text (although present in methods). This should be also included in legend.

Reviewer #2 (Remarks to the Author):

Ma et al performed a genome-scale CRISPR/Cas9 KO screen using a chimeric alphavirus that expresses structural proteins from EEEV to identify additional uncharacterized entry factors. They identified LDLR as a candidate receptor, showing that its presence facilitates binding and infection by multiple alphaviruses. Furthermore, the authors revealed a low-affinity interaction with LA domain 3 (LA3), which could be enhanced by concatenated LA3 sequences. Notably, soluble decoy proteins containing multiple LA3 repeats could effectively inhibit EEEV infection in cell culture and in mice. Overall, the findings convincingly show that LDLR is a low-affinity receptor for various alphaviruses and suggest a potential avenue for developing antivirals. It is a well-written manuscript which sheds light on the flexible use of alphaviruses on the family of LDLR and LDLR-like receptors. The findings are highly significant and the authors claims are firmly supported by the data. I have only a few minor suggestions are included.

1. The inset of figure 3e is not helping to show the results better. It is hard to tell the colocalization of LDLR and WGA, zoom in more of both LDLR and WGA (split and overlap) may demonstrate the data better.

2. Extended Data Figure 2:

Extended Data Figure 2c-e are not correctly ordered, including the figure legend and quotation in the main text, check the paragraph starting from line 170, e.g., switch c and e; line 185: there is no relevance to Extended Data Figure 2c.

3. Line 232-233: incomplete bracket

4. Line 309: f and g

5. Line 341: LDLR-Fc (a), LBD-Fc, (a) or LA3-Fc (b) to LDLR-Fc (a), LBD-Fc (a), or LA3-Fc (b)

REVIEWER COMMENTS

Reviewer #1:

This manuscript uses a variety of sophisticated approaches to report the identification of a new cellular entry receptor (LDLR) for specific members of the alphavirus family. Alphaviruses include many medically important pathogens, for which there are no licensed anti-virals or medicines. Overall the studies are multifaceted and well done. The authors make use of a comprehensive CRISPR-based screen in the neuroblastoma cell line N2a to identify LDLR and confirms its role through a variety of approaches. Viral attachment and internalisation studies were then performed using the monocytic cell line THP-1, which was engineered to over express LDLR. Furthermore, LDLR is shown convincingly to bind directly to VEEV using biolayer interferometry, and the precise binding site more specifically mapped to specific amino acids in the LA3 domain. Despite the low affinity nature of LA3 interactions with virus, the authors define a decoy composed of tandem LA3 repeat domains that limits virus infection of THP-1 cells over expressing LDLR. Their in vivo data is limited but preliminarily suggests a role for LA3-VEEV interactions. Together, these results, if further validated using the suggested changes below, should constitute an important advance in our understanding of alphavirus biology.

We appreciate the favorable and supportive summary.

Major changes

Figure 2 and 4. Considering that the majority of data in figure 1 was established using N2a cells that express LDLR without experimental manipulation, it is not clear why studies in this figure 2 and 4 were not also performed in N2a cells +/- LDLR. These should ideally been done and compared, especially as LDLR was over expressed in THP-1 cells and may not be particularly representative. This is also the case for Figure 4, in which LD3 tandem repeats were used to suppress infection of THP-1 cells. Data generated using a cell type with endogenous LDLR expression are required.

*We used THP-1 cells because at baseline they express low levels of LDLR (or other alphavirus receptors), and thus allowed a clean system to see effects on binding, internalization, and analysis of decoy receptor neutralization. Notwithstanding this point, we agree with the Reviewer. We have also added binding and internalization in LDLR KO and complemented N2a cells (**Extended Data Fig 2a-b**). For technical reasons, instead of using the N2a cells for neutralization studies, we have provided LA3-COMP inhibition data with SH-SY5Y cells, a different neuroblastoma cell line (new **Extended Data Fig 4d**).*

The majority of studies using cells with endogenous LDLR expression were undertaken using N2a cells, a neuroblastoma cell line that is most similar to immature neuronal precursors found in vivo. It is not clear how relevant these findings are for their in vivo non-immortalised neuronal counterparts, which often lack expression of genes detected in less differentiated cell lines. The maturity status of neurons has a defining impact of cell susceptibility to alphavirus infection (e.g. see extensive studies on neuron maturity and SFV/SINV infection). The relative immaturity of N2a cells may make them poor representatives of most neuron cell types found in the developed brain. In the absence of CNS specific in vivo data, the authors should define expression of LDLR in different neurons and discuss these limitations.

*We agree with this comment. Using an available Mouse Cell Atlas database (<https://bis.zju.edu.cn/MCA/>), we determined that *Ldlr* mRNA is abundantly expressed in mouse*

brain in oligodendrocytes, microglia, astrocyte, but interestingly, not in neurons (Extended Data Fig 5). We have also added a short discussion of this in the Results section (p. 10).

Figure 4i,j. The in vivo data is limited and incomplete. It is not clear whether the in vivo phenotype induced by LA3(5)-COMP administration is CNS specific and e.g. due to interference specifically with neuronal infection by virus. A better description of the clinical phenotypes of mice succumbing (or reaching clinically defined end points) is necessary, as is the determination of virus dissemination to target tissues (e.g. lymphoid tissue and brain), with or without LA3(5)-COMP administration.

We agree with this comment as well. One note is the Madariaga virus (MADV) infection and pathogenesis model in C57BL/6 mice is somewhat unique; beyond weight loss, the animals do not show much of a prodrome of infection (with neurological signs or other phenotypes) but instead just rapidly succumb within a few days period. To address this concern, we instead performed a time course of viral load analysis and have included data showing lower levels of MADV infection in several tissues including the brain after LA3(5)-COMP treatment (Fig 4k-l).

Minor changes:

Lines 42-43. It would be better to additionally define SFV as a member of the old world alphaviruses, especially as it is referred to later in the introduction and is one of the viruses studied here.

We have added SFV as suggested.

Lines 92-96. Clarify how substituting non-structural genes from EEEV (BSL3) with those from SINV (BSL2) makes the chimera suitable for work at BSL2. Is the assumption that virulence of EEEV depends on non-structural gene expression? Please clarify and reference as appropriate.

We have added a comment that SINV chimerization substantially decreases the virulence of encephalitic alphaviruses (p. 5).

Figure 2. All timing information of these assays is missing from legend and main text (although present in methods). This should be also included in legend.

We have added this to the legend and text as requested in the main text and Legend.

Reviewer #2:

Ma et al performed a genome-scale CRISPR/Cas9 KO screen using a chimeric alphavirus that expresses structural proteins from EEEV to identify additional uncharacterized entry factors. They identified LDLR as a candidate receptor, showing that its presence facilitates binding and infection by multiple alphaviruses. Furthermore, the authors revealed a low-affinity interaction with LA domain 3 (LA3), which could be enhanced by concatenated LA3 sequences. Notably, soluble decoy proteins containing multiple LA3 repeats could effectively inhibit EEEV infection in cell culture and in mice. Overall, the findings convincingly show that LDLR is a low-affinity receptor for various alphaviruses and suggest a potential avenue for developing antivirals. It is a well-written manuscript which sheds light on the flexible use of alphaviruses on the family of LDLR and LDLR-like receptors. The findings are highly significant and the authors claims are firmly supported by the data. I have only a few minor suggestions are included.

We appreciate the positive comments.

1. The inset of figure 3e is not helping to show the results better. It is hard to tell the colocalization of LDLR and WGA, zoom in more of both LDLR and WGA (split and overlap) may demonstrate the data better.

*We agree and have added a high magnification inset to better show the overlap. We also added the FITC channel to highlight the distribution of LDLR on the surface on THP-1 cells (**Extended Data Fig 3b**).*

2. Extended Data Figure 2: Extended Data Figure 2c-e are not correctly ordered, including the figure legend and quotation in the main text, check the paragraph starting from line 170, e.g., switch c and e; line 185: there is no relevance to Extended Data Figure 2c.

*We apologize that the Figure panel was not in the correct order. We have swapped the letters to match the main text and legend (now new **Extended Data Fig 3**).*

3. Line 232-233: incomplete bracket

We have corrected this error.

4. Line 309: f and g.

We have corrected this error.

5. Line 341: LDLR-Fc (a), LBD-Fc, (a) or LA3-Fc (b) to LDLR-Fc (a), LBD-Fc (a), or LA3-Fc (b)

We have corrected this error.

REVIEWERS' COMMENTS

Reviewer #1 (Remarks to the Author):

The authors have gone to considerable lengths to improve their manuscript. All comments have been well rebutted and requested changes have been satisfactorily undertaken.

REVIEWERS' COMMENTS

Reviewer #1:

The authors have gone to considerable lengths to improve their manuscript. All comments have been well rebutted and requested changes have been satisfactorily undertaken.

We appreciate the supportive comment.